# Revealing core-valence interactions in solution with femtosecond X-ray pump X-ray probe spectroscopy

Robert B. Weakly [1], Chelsea E. Liekhus-Schmaltz[1], Benjamin I. Poulter [1], Elisa Biasin [2,3], Roberto Alonso-Mori [4], Andrew Aquila [4], Sébastien Boutet [4], Franklin D. Fuller[4], Phay J. Ho [5], Thomas Kroll[6], Caroline M. Loe[1], Alberto Lutman [4], Diling Zhu[4], Uwe Bergmann [7], Robert W. Schoenlein [2,4], Niranjan Govind [3] & Munira Khalil [1] ✉

Femtosecond pump-probe spectroscopy using ultrafast optical and infrared pulses has become an essential tool to discover and understand complex electronic and structural dynamics in solvated molecular, biological, and material systems. Here we report the experimental realization of an ultrafast two-color X-ray pump X-ray probe transient absorption experiment performed in solution. A 10 fs X-ray pump pulse creates a localized excitation by removing a $1s$ electron from an Fe atom in solvated ferro- and ferricyanide complexes. Following the ensuing Auger–Meitner cascade, the second X-ray pulse probes the Fe $1s \rightarrow 3p$ transitions in resultant novel core-excited electronic states. Careful comparison of the experimental spectra with theory, extracts +2 eV shifts in transition energies per valence hole, providing insight into correlated interactions of valence $3d$ with $3p$ and deeper-lying electrons. Such information is essential for accurate modeling and predictive synthesis of transition metal complexes relevant for applications ranging from catalysis to information storage technology. This study demonstrates the experimental realization of the scientific opportunities possible with the continued development of multicolor multi-pulse X-ray spectroscopy to study electronic correlations in complex condensed phase systems.

Femtosecond (fs) pump–probe spectroscopy is now used extensively in the optical and infrared (IR) regimes to understand complex chemical phenomena in the condensed phase following the development of high intensity, commercially available, and tunable ultrafast laser systems. In these widely used experiments, the pump pulse initiates a non-equilibrium process in the excited-state (or ground-state) of the system of interest, and the delayed probe pulse monitors the time

evolution of the system. For example, optical transient spectroscopy experiments measure signatures of evolving electronic states following photoexcitation in complex systems[1,2]. On the other hand, transient IR experiments measure structural dynamics and non-equilibrium vibrational relaxation in photo-excited systems[3,4]. Femtosecond pump–probe spectroscopy in the condensed phase has resulted in new discoveries in the fields of biology, chemistry, and

[1]Department of Chemistry, University of Washington, Seattle, WA 98195, USA. [2]Stanford PULSE Institute, SLAC National Accelerator Laboratory, Menlo Park, CA 94025, USA. [3]Physical and Computational Sciences Directorate, Pacific Northwest National Laboratory, Richland, WA 99352, USA. [4]Linac Coherent Light Source, SLAC National Accelerator Laboratory, Menlo Park, CA 94025, USA. [5]Chemical Sciences and Engineering Division, Argonne National Laboratory, Lemont, IL 60439, USA. [6]Stanford Synchrotron Radiation Light Source, SLAC National Accelerator Laboratory, Menlo Park, CA 94025, USA. [7]Department of Physics, University of Wisconsin-Madison, Madison, WI 53706, USA. ✉e-mail: mkhalil@uw.edu

material science. The pump–probe experiments have served as precursors to coherent nonlinear multidimensional techniques that are now used extensively to map charge and energy transfer pathways in complex systems[5,6]. An important limitation, however, is that fs optical and IR spectra are only indirectly sensitive to valence charge distributions across specific atomic sites.

The advent of third generation synchrotrons and the development of table-top X-ray sources has resulted in the development of transient X-ray techniques for measuring the X-ray absorption and emission spectra of core electrons in solvated complexes following photoexcitation with an optical pump pulse[7–11]. X-ray spectroscopy is an element-specific probe of electronic and atomic structure and fs X-ray pulses from X-ray free electron lasers (XFELs) are allowing researchers to routinely measure X-ray absorption and emission spectra of ultrafast photochemical processes in solution[12–16]. Optical pump X-ray probe experiments have also enabled researchers to measure coherently coupled electronic and atomic motions and ultrafast electron delocalization in complex photochemical phenomena in solution[17–20].

The generation of tunable, high intensity, time-delayed, femtosecond X-ray pulse pairs has been recently demonstrated at various XFELs around the world[21–25]. These technological developments have enabled X-ray pump X-ray probe experiments studying nuclear and electronic dynamics at different atomic sites in small molecules in the gas phase through various electron ionization detection schemes[26–28]. Two-pulse X-ray photon correlation techniques have measured non-equilibrium structure correlations on short length scales in solutions and solids[29–35]. Additionally, nonlinear light matter interactions with intense X-ray pulses have resulted in seeded stimulated emission signals at the Mn K-edge in concentrated solutions[36].

Recently, we proposed and theoretically modeled a novel approach to measure core-valence electronic correlations in solvated chemical systems via ultrafast X-ray pump X-ray probe (XPXP) transient absorption spectroscopy, using the transition metal complex, $K_4Fe^{II}(CN)_6$, as a model system[37]. X-ray pump pulses were used to create a localized (element-specific) excitation consisting of a $1s$ core hole in the Fe atom. We used a combination of atomic electron cascade calculations and excited-state time-dependent density functional theory (TDDFT) calculations to predict changes in the X-ray probe transmission near the Fe K-edge following the X-ray pump interaction. Our work found several spectral features below the Fe K-edge absorption edge that reported on the ligand-field splitting and chemically relevant $3p$ and $3d$ electron interactions.

This paper is the experimental realization of our previous theoretical work described above[37], presenting an XPXP transient absorption experiment of molecules in solution. The measured XPXP transient absorption spectra combined with simulations directly measure the oxidation state dependent electronic cascade pathways and the chemically relevant $3p$–$3d$ valence interaction strengths in $Fe^{II}$ and $Fe^{III}$ hexacyanoferrates dissolved in water. This study also serves as an experimental demonstration of the feasibility of two-color XPXP transient absorption, paving the way for multi-pulse nonlinear multidimensional X-ray spectroscopy in solution.

## Results

### Implementation of the XPXP experiment

Figure 1a depicts the generation of an energy tunable ~10 fs X-ray pulse pair, separated by a time delay ($\tau$) coincident on a thin liquid jet (250 μm) containing an aqueous solution of either $K_4Fe^{II}(CN)_6$ or $K_3Fe^{III}(CN)_6$[38]. The X-ray pump pulse (7.2 keV, blue dashed line in Fig. 1c) ionizes the sample by removing a $1s$ electron from the Fe atom. The energy of the X-ray pump pulse is chosen to ensure that fluctuations in its photon energy or spectral shape do not influence the character of the initially prepared ionized state in the Fe complex. Figure 1b provides an example of an Auger-Meitner electron cascade

and we stress that this is only one of a multitude of possible pathways explored by the system following the interaction with the X-ray pump pulse. The representative cascade pathway in Fig. 1b proceeds in experimental time as follows: (1) a pump photon ejects a $1s$ electron to the continuum, (2) a Kα fluorescence event fills the $1s$ hole and emits a photon, (3) an Auger–Meitner decay fills the new $2p$ hole while removing a $3p$ electron, (4) Coster–Kronig decay fills the $3s$ hole, ejecting a $3dt_{2g}$ electron, resulting in an electron configuration of [Ne] $3s^23p^{x=5}3dt_{2g}^{y=4}$. Following the Auger–Meitner electronic cascade (5), the second X-ray pulse centered at 7.06 keV probes the ensemble of resultant core-electronic excited states containing $3p$ holes. The transmitted X-ray probe pulse is spectrally resolved to measure the transient X-ray absorption spectrum (see Fig. 1a, c).

### Prediction of transient core-excited electronic states probed in the XPXP experiment

To predict the transient core-excited electronic states created in the X-ray probe's spectral window (7060 ± 10 eV) following the removal of a $1s$ core electron in the Fe atom by the X-ray pump pulse, we performed an electron cascade Markov-chain Monte Carlo (MC-MC) calculation, as described previously[37]. These calculations serve as a starting point for understanding the spectral features observed in this XPXP experiment. Figure 2 shows the time evolution of [Ne] $3s^23p^53dt_{2g}^y$ electronic configurations in the $Fe^{II}$ (solid lines) and $Fe^{III}$ (dashed lines) atomic systems produced by the MC-MC simulations in the first 100 fs after ionization of a $1s$ electron. These electronic states are the most probable (~20%) among all states populated at the time-scale probed in our experiment. Identically colored lines represent states that have lost an equivalent number of electrons from the valence through Auger–Meitner events ($N_{Auger}$) as indicated in the legend of Figure 2. For example, the solid and dashed blue lines ($N_{Auger}$=0) in Fig. 2 correspond to the initial $3d^6$ and $3d^5$ configurations for $Fe^{II}$ and $Fe^{III}$ atoms, respectively. Interestingly, we note that the probability of observing the [Ne]$3s^23p^53dt_{2g}^6$ state of the $Fe^{II}$ system decays to zero within 1 fs. Given the experimental X-ray pulse durations of ~10 fs, we expect the largest contributions from $3p^5$ core-excited states with 0, 1, and 3 Auger–Meitner events ($N_{Auger}$ = 0, 1, and 3) depending on the starting oxidation state of the Fe atom. The relative contributions of each of the core-excited states to the measured XPXP transient absorption spectrum depend on their calculated probabilities plotted in Fig. 2. The probabilities in Fig. 2 are on the same order of magnitude as those calculated from experimentally determined fluorescence yields of the Fe atom[39], and discussed in Supplementary Note 4. The MC-MC simulations also predict the average oxidation state during the first 10 fs of the electronic cascade to be $3d^{4.2}$ and $3d^{3.8}$ for the $Fe^{II}$ and $Fe^{III}$ systems, respectively (see Supplementary Note 3). Combining all the information gleaned from the electron cascade calculations, we would expect to observe the following in the transient XPXP signal: (i) two or three spectral features representing $1s \rightarrow 3p$ transitions to distinct electronic states for the $Fe^{II}$ and $Fe^{III}$ samples, and (ii) a lack of the $1s \rightarrow$ [Ne]$3s^23p^53dt_{2g}^6$ transition in the $Fe^{II}$ data.

### Femtosecond XPXP signal measures core-valence interactions

Figure 3 displays the XPXP signal as a change in transmission of the X-ray probe spectrum following $1s$ excitation of the Fe atom in aqueous solutions of $K_4Fe^{II}(CN)_6$ or $K_3Fe^{III}(CN)_6$ complexes by the X-ray pump pulse. The data are measured at a nominal delay time of 0 fs between the two 10 fs X-ray pulses. Both plots display the presence of transient $1s \rightarrow 3p$ absorption features as two negative spectral features, blue shifted from the molecular Fe Kβ energy (7058 eV and 7059 eV for $Fe^{II}$ and $Fe^{III}$ molecular complexes respectively, vertical dashed lines in Fig. 3). These transitions are a result of new core-excited electronic states with $3p$ holes formed following the initial steps of an Auger–Meitner cascade (see Fig. 1b for an example of electron

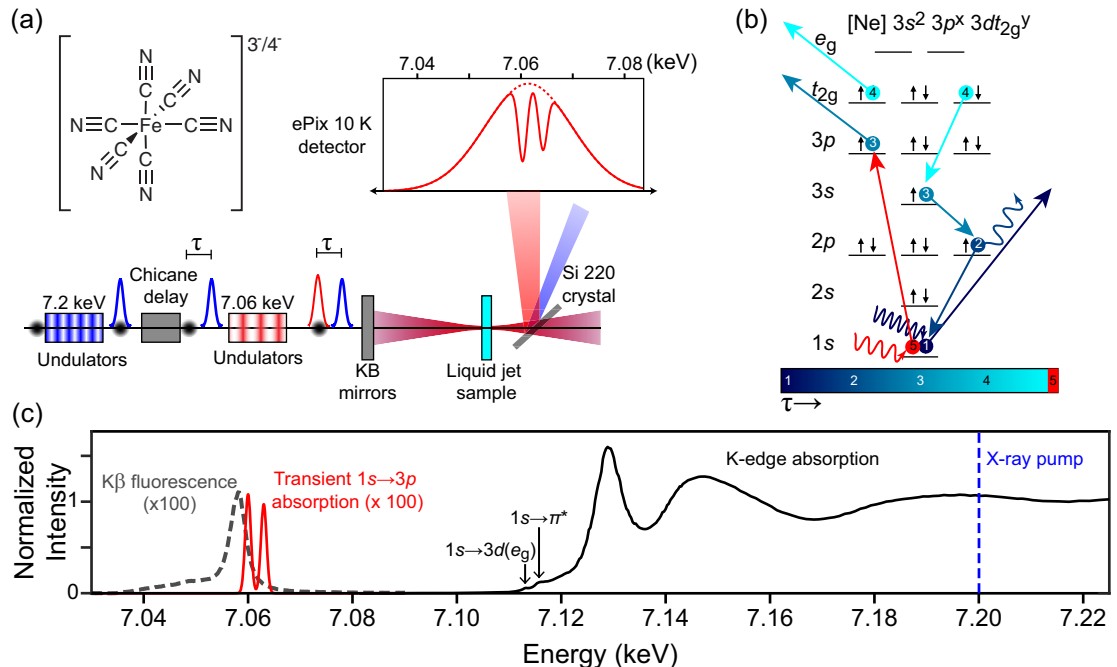

**Fig. 1 | Overview of the XPXP experiment. a** Experimental layout. Both the 7.20 keV X-ray pump (blue) and the 7.060 keV X-ray probe (red) are generated collinearly by separate undulators from the same electron bunch (black circle). Insertion of a chicane delay allows for the pump–probe timing ($\tau$) to be controlled. Both X-ray pulses are focused onto the thin liquid jet by a pair of Kirkpatrick–Baez (KB) mirrors to a spot size of 100 nm. The transmitted X-ray spectra are measured using an analyzer crystal and an X-ray detector (ePix 10 K). **b** Example of an Auger–Meitner cascade in the iron complexes following removal of an Fe 1$s$ electron. The example shows that the 1$s$ hole is quickly filled by various processes (labelled 2–4) including fluorescence, Auger–Meitner decay, and Coster–Kronig decay events. The resultant electronic states have an electronic configuration of

[Ne]3$s^2$3$p^x$3$dt_{2g}^y$. After delay time, $\tau$, the X-ray probe pulse, resonant with the 1$s \rightarrow 3p$ transitions interrogates the core electronic excited states formed following the X-ray pump interaction. **c** Equilibrium ground state Fe K-edge X-ray absorption spectrum and K$\beta$ X-ray emission spectrum of a 500 mM aqueous solution of K$_4$Fe$^{II}$(CN)$_6$. The X-ray absorption reference spectrum (solid black, normalized to the post-edge) displays weak pre-edge features of 1$s \rightarrow 3de_g$ and 1$s \rightarrow \pi^*$ transitions characteristic of an Fe(II) complex as described in ref. 38. Transient 3$p$ absorption features measured by the X-ray probe pulse are shown in red. See Supplementary Note 4 for further details on the treatment of the X-ray absorption, emission spectra, and transient signals.

cascade). The XPXP data were processed by separately averaging and subtracting the X-ray pumped and unpumped probe spectra to produce the change in transmission ($\Delta T/T$) data and error bounds in Fig. 3. Individual X-ray shots exhibiting abnormal characteristics in intensity or spectral distribution were excluded from the analysis. Details on the processing of the raw data are provided in Supplementary Note 2. The XPXP signal is dependent on the X-ray pump and probe pulse fluence and the XPXP transient absorption signal is absent in pure water (see Supplementary Figs. 3 and 6).

The spectral features in the XPXP signal are fit using Gaussian lineshapes. The best fit is plotted as solid lines in Fig. 3a, b, and the peak amplitudes, positions and linewidths extracted from the fit are listed in Supplementary Table 2. From the fits, we determine that the XPXP transient signals in Fe$^{II}$ and Fe$^{III}$ complexes exhibit two 1$s \rightarrow 3p$ transitions at ~7060 and ~7062 eV. The peaks in the transient difference XPXP spectra from the Fe$^{II}$ and Fe$^{III}$ complexes are blue-shifted from the peak of the K$\beta$ X-ray emission signals (Supplementary Fig. 8) by 1.9 and 1.2 eV respectively. The fits reveal that the overall transient signal in the Fe$^{II}$ complex is blue shifted by 0.35 eV with respect to the Fe$^{III}$ signal. We note that this shift is within the 0.4 eV resolution of the detection spectrometer. The widths of all the 1$s \rightarrow 3p$ transmission peaks in the transient XPXP spectra for both samples are ~2 eV. These widths are determined by the non-radiative lifetime of the 3$p$ hole and are narrower compared to the width of the K$\beta$ emission peak, which is determined by the shorter lifetime of the 1$s$ hole. The narrower peaks in the XPXP spectra make it easier to resolve individual core-excited electronic states. We compare the intensities of the spectral features by their integrated peak areas (Supplementary Table 2) and find that for the Fe$^{II}$ sample, the peak at ~7060 eV is 15% more intense than the

peak at ~7062 eV. In the case of the Fe$^{III}$ sample, the peak at ~7060 eV is 3% more intense than the peak at ~7062 eV. Assuming similar dipole strengths for all 1$s \rightarrow 3p$ transitions, the integrated area of each peak corresponds to the relative population of that particular 3$p$ state in the sample of interest. We also observe that the transitions in the XPXP Fe$^{III}$ data show overall greater intensity by 10% of the observed transitions relative to the Fe$^{II}$ data and we attribute this to an initial additional hole in the 3$dt_{2g}$ orbital. Our observations are in agreement with previous calculations on the generalized case of 3$p$ vacancy dependent M-edge spectroscopy[40]. In summary, the 1$s \rightarrow 3p$ XPXP transient absorption spectra for the Fe$^{II}$ and Fe$^{III}$ complexes show remarkable similarities in the peak positions, integrated areas, and lineshapes.

To aid the interpretation of the data shown in Fig. 3, we perform TDDFT calculations of 1$s \rightarrow 3p$ X-ray absorption spectra for a variety of reference electronic configurations on separate geometries for both Fe$^{II}$ and Fe$^{III}$ compounds following our previously published computational approach[37]. Additional computational details are provided in Supplementary Note 5. Figure 4a, b show calculated 1$s \rightarrow 3p$ transitions in core-excited states with specific electronic configurations. The TDDFT calculation reports that each progressive hole in the 3$d$ valence shell produces a corresponding shift in the 1$s \rightarrow 3p$ energy gap. From Fig. 4, we note that in the [Ne]3$s^2$3$p^5$3$dt_{2g}^y$ configurations, each additional hole in the $t_{2g}$ orbital corresponds to a + 2 eV shift of the 1$s \rightarrow 3p$ calculated transition. This shift serves as an indirect reporter on the strength of the 3$p$–3$d$ interactions in the molecular complex under investigation. The TDDFT calculations reveal that 1$s \rightarrow 3p$ transitions for states with electronic configurations of the type [Ne]3$s^2$3$p^{x<5}$3$dt_{2g}^y$ are shifted ~6 eV to the blue of the experimental K$\beta$ emission peak. Further, we see that the peak from the 1$s \rightarrow$ [Ne]3$s^2$3$p^5$3$dt_{2g}^6$

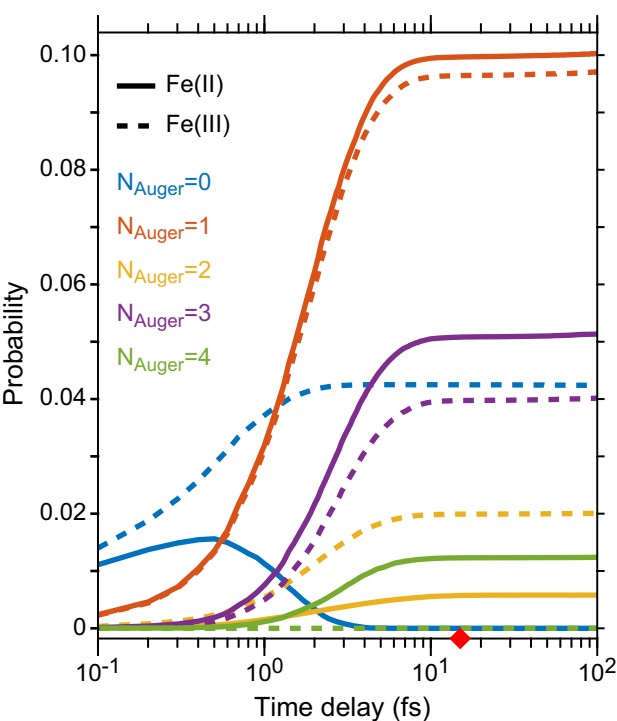

**Fig. 2 | Time-dependent probabilities of $[Ne]3s^23p^53dt_{2g}^y$ electronic states following $1s$ hole generation.** Monte Carlo electron cascade calculations of atomic $Fe^{II}$ (solid lines) and $Fe^{III}$ (dashed lines) are shown above where the color corresponds to an equivalent number of electrons lost from the valence through Auger–Meitner events ($N_{Auger}$) in the two Fe species. The probabilities for each state are given with respect to all electronic configurations (including non $[Ne]3s^23p^5$ states). At long times ($\geq 1$ ps), $[Ne]3s^23p^53dt_{2g}^y$ states comprise 18 and 21% of all possible electronic states for $Fe^{II}$ and $Fe^{III}$ atoms, respectively. The instrument response function of the present experiment (-14 fs) is denoted by the red diamond on the time axis. Source data are provided as a Source data file.

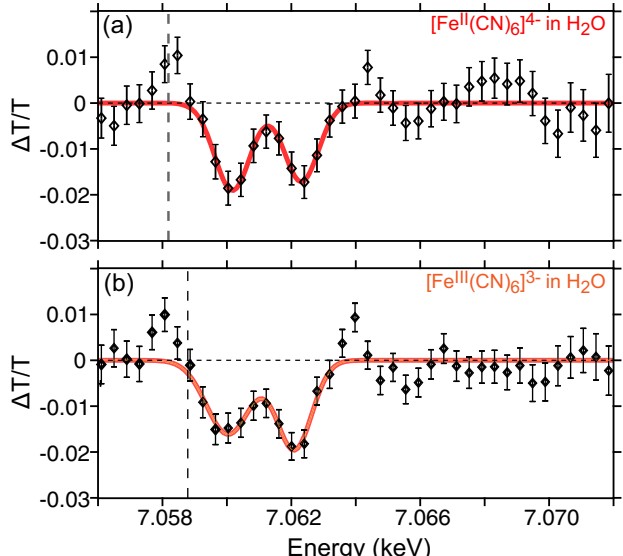

**Fig. 3 | Experimental XPXP Transient Absorption Spectra.** XPXP signal plotted as change in transmission of the X-ray probe ($\Delta T/T$) following X-ray pumping of the aqueous solutions of (**a**) $K_4Fe^{II}(CN)_6$ and (**b**) $K_3Fe^{III}(CN)_6$ measured at $\tau = 0$ fs. Negative peaks centered at -7.060 and 7.062 keV correspond to additional $1s \rightarrow 3p$ absorption due to the presence of $3p$ holes resulting from the Auger–Meitner cascade, reducing the intensity of the X-ray probe pulse through the sample. The transient data is fit (red line) using a sum of two Gaussians and best fit values are listed in Supplementary Table 2. The vertical dashed line represents the molecular $K\beta$ emission line. Error bars are calculated as the standard error for each energy bin. See Supplementary Note 2 for more details. Source data are provided as a Source data file.

configuration for the $Fe^{II}$ species (starred in Fig. 4a), would be located around the peak of the $K\beta$ emission line.

Combining the electron cascade (Fig. 2) and the TDDFT (Fig. 4) calculations, we assign the peaks at 7060 and -7062 eV in the $Fe^{II}$ and $Fe^{III}$ XPXP transient absorption spectra (Fig. 3) to $1s \rightarrow [Ne]3s^23p^53dt_{2g}^5$ and $1s \rightarrow [Ne]3s^23p^53dt_{2g}^4$ transitions, respectively. Both the MC-MC electron cascade simulations and the TDDFT calculations confirm that the valence-hole-free state in the $Fe^{II}$ complex does not survive past the nominal $1s$ core lifetime, and the observed transitions in both complexes originate from similar core-excited states. From the integrated area of the peaks in the XPXP data, we observe that the electron cascade in the $Fe^{III}$ complex generates more highly ionized states in the relative populations of $t_{2g}^5$ and $t_{2g}^4$ states compared to the electron cascade in the $Fe^{II}$ complex. A comparison of the TDDFT calculations and the XPXP experimental data reveals that excited states with configurations of $[Ne]3s^23p^{x<5}3dt_{2g}^y$ are not formed in this experiment, confirmed by the lack of observed transitions above the signal to noise in Fig. 3 for X-ray probe energies greater than 7063 eV. We note that several factors could contribute to the lack of observing $3p^{x<5}$ holes in this XPXP experiment including (i) data collection at zero nominal delay between the X-ray pump and probe pulses, limiting the timeframe for electron cascade evolution, and (ii) contribution to the electron cascade by the surrounding ligand or solvent electrons resulting in an effective quenching of the solute's electron cascade signal by decreasing the $3p^{x<5}$ lifetime. We consider the possibility that the peaks seen in the XPXP experimental signal could arise from the splitting of the $3p^5$ configurations via spin–orbit interactions. In reference calculations of $3p^5$ configurations, the $3p$ spin–orbit

coupling was computed to be -1.3 eV for both $Fe^{II}$ and $Fe^{III}$ complexes, in agreement with previously published data on transition metal ions[41]. Given that the $3p$ spin–orbit coupling is less than the linewidth and the energy separation of the observed peaks in the XPXP transient spectra shown in Fig. 3, we confirm the assignment of the observed peaks to $1s \rightarrow [Ne]3p^53dt_{2g}^5$ and $1s \rightarrow [Ne]3p^53dt_{2g}^4$ transitions in solvated $Fe^{II}$ and $Fe^{III}$ complexes. Our combined femtosecond XPXP experimental data and simulation protocol provide a direct experimental measure of the -2 eV blue-shift of the $1s \rightarrow 3p$ dipole transition as a function of the number of $3d$ holes in solvated Fe complexes.

## Discussion

Measuring the time-evolution of core–valence interactions in transition metal complexes is crucial for controlling electronic correlations in molecules and materials being developed for catalytic, magnetic, and information storage applications. The XPXP experiment reported here is sensitive to the $3p$–$3d$ Coulomb and valence interactions as a function of the electronic configuration, oxidation, and spin state of the Fe atom in solvated molecules. By directly probing core-to-core, dipole-allowed, $1s \rightarrow 3p$ transitions, our data reports on both the $1s$ relaxation and the $3p$–$3d$ interactions (Supplementary Fig. 10).

In the novel core-excited electronic states produced by the X-ray pump, core and valence electrons rearrange and relax due to the constantly changing electrostatic shielding in the Fe atom, which includes shifts from $3p$–$3d$ and $3d$–$3d$ Coulomb and valence exchange interactions, crystal field interactions and spin–orbit interactions. These interactions are encoded in numerous transitions connecting the typical K- ($1s$), L- ($2p$), or M- ($3p$) edges via X-ray absorption and emission spectra of transition metal complexes measured at synchrotrons, XFELs and with table-top HHG-based sources (see Fig. 5)[42]. At the K-edge, $K\beta$ fluorescence XES probes the $3p$–$3d$ exchange energy through the relative intensities of the spectral features sensitive to spin and $3d$ occupancy[43–45]. Similarly, L-edge spectroscopy has been used extensively to monitor the covalency, spin, and back-bonding

in transition metals[46–49] and M-edge spectra directly measure $3p \rightarrow 3d$ transitions[50]. The extraction of core electronic state-specific information from equilibrium and optically-pumped X-ray absorption spectroscopy (XAS) and XES spectral features at the K-, L- and M-edges

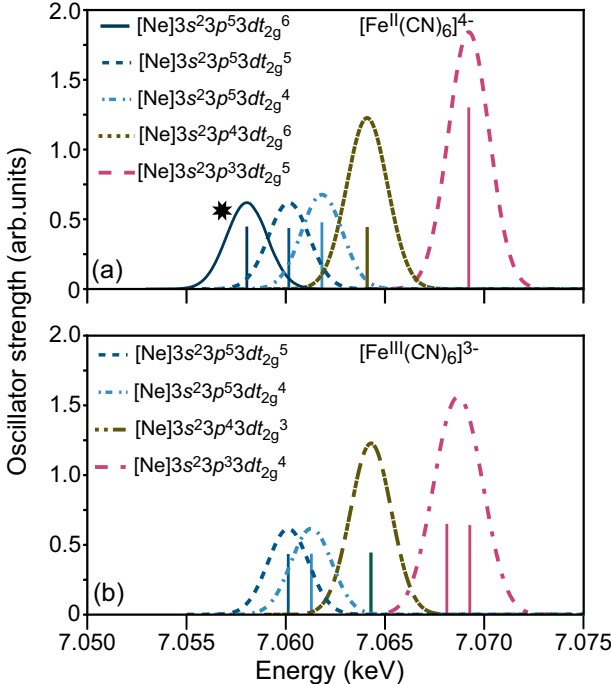

**Fig. 4 | TDDFT calculations of $1s \rightarrow 3p$ X-ray absorption spectra.** Calculated shifts in the $1s \rightarrow 3p$ transition energy and oscillator strengths for (**a**) [Fe$^{II}$(CN)$_6$]$^{4-}$ and (**b**) [Fe$^{III}$(CN)$_6$]$^{3-}$, for a selected sample of core-excited electronic states. The starred transition is not observed in the data. Each of the calculated transitions are broadened by 1.5 eV and shifted by 143.8 eV to match the experimental $1s \rightarrow 3de_g$ pre-edge feature in the Fe K-edge X-ray absorption spectrum of K$_4$Fe$^{II}$(CN)$_6$ (plotted in ref. 38).

is limited given the lifetime broadening of 1s and 2p core holes[51] and or further complicated by many possible dipole-allowed transitions to a dense continuum of valence states. Furthermore, the temporal resolution of transient XAS is limited by the lifetime of the valence excited state and the pulse duration of the optical pump pulse. In contrast, the time-resolution of XPXP experiments can be shorter given the availability of few fs X-ray pulses at XFELs.

In this XPXP study, our ability to extract microscopic information from the $1s \rightarrow 3p$ spectral features is theoretically limited by the 3p non-radiative lifetime broadening and experimentally by the spectrometer (0.40 eV)[52] resolution and the 10 fs temporal widths of the X-ray pump and X-ray probe pulses. Given that the temporal profiles of the X-ray pulses are much longer than the core-hole-lifetime of the 1s electron ionized by the X-ray pump pulse, the transient XPXP absorption spectra probe a near static population of core-excited states with varying 3d holes. With the generation of tunable attosecond hard X-ray pump and probe pulses, the XPXP experiment described here could measure transient absorption spectra prior to Auger−Meitner decay and would be uniquely sensitive to tracking specific core-excited electronic states, time-dependent 3p−3d electronic correlations, and the time-evolution of pure valence electronic coherences with atomic specificity.

The interpretation of the experimental XPXP transient absorption spectra in this study relies on the MC-MC simulation of the electron cascade in isolated Fe$^{II}$ and Fe$^{III}$ atoms to model the effect of the X-ray pump pulse and the TDDFT calculations of the $1s \rightarrow 3p^5$ transitions for a select group of core-excited states in [Fe$^{II}$(CN)$_6$]$^{4-}$ and [Fe$^{III}$(CN)$_6$]$^{3-}$ complexes. Despite the atomic nature of the electron cascade calculations and the single excitation nature of TDDFT calculations, we are able to predict and measure core hole relaxation manifesting in 3p and 3d transition energies as a function of the 3d hole density. We stress that the successful demonstration of a fs XPXP experiment of a solvated molecular system, as shown here, increases the urgency of developing theoretical tools to accurately model multi-pulse X-ray-matter interactions with complex molecules in solution. Such calculations will be crucial for understanding how electronic correlations, spin−orbit, ligand−field, and solute-solvent interactions

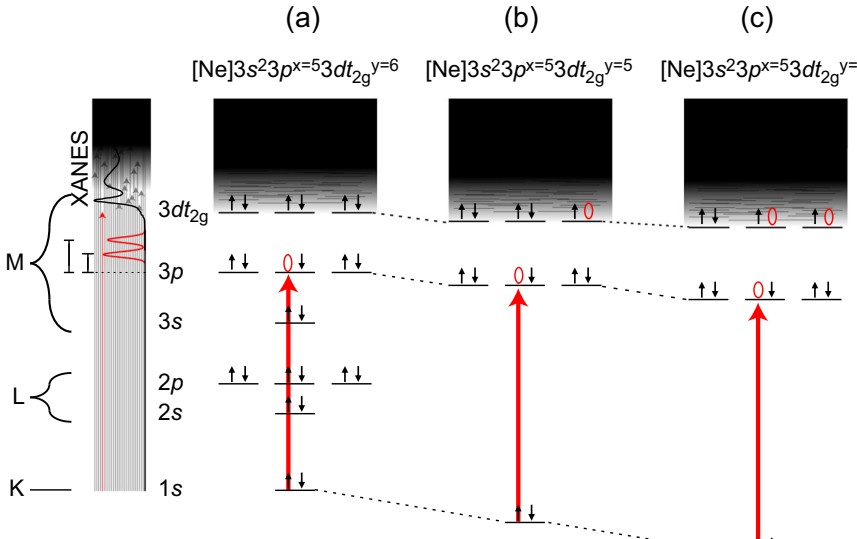

**Fig. 5 | Comparison of the XPXP transient absorption experiment to an optically pumped transient XAS experiment.** The thick red arrows correspond to dipole allowed transitions in the XPXP experiment and the thin red line represents the quadrupolar transitions probed in a transient XAS experiment. The red ovals indicate holes. XPXP transient absorption spectra measure the progressive blue-shifting of the $1s \rightarrow 3p$ transition energy resulting from the unequal stabilization of 1s, 3p and 3d orbitals as a function of the number of holes in the $t_{2g}$ level. Here we probe core-excited 3p states with filled (**a**) and unfilled (**b**) and (**c**) $3dt_{2g}$ shells and the resultant peaks are spectrally isolated around the Kβ emission energy peak and well-separated from the absorption edge.

are manifested in XPXP transient absorption spectra of core-excited molecular complexes in solution.

Along with theoretical developments, experimental developments in generating intense, multi-color, time-delayed, attosecond pulse-pairs in the hard X-ray regime will result in an extension of the fs XPXP spectroscopy presented here to a coherent multidimensional nonlinear X-ray experiment on molecular complexes in solution. An analogy is the development of third-order nonlinear coherent multidimensional optical and IR spectroscopy to measure couplings between excitonic states and anharmonic vibrations, respectively, following the establishment of femtosecond optical pump–probe and IR pump–probe and transient grating experiments. Mukamel and co-workers have proposed several coherent multidimensional techniques in the X-ray regime to elucidate coherent electronic charge and energy transfer pathways[53–55]. With the availability of tunable attosecond hard X-ray pulses, it will be possible to create and collapse coherences within the $1s$ core-hole-lifetime, generating opportunities for investigating the coherences between specific core-excited electronic states of complex systems in solution.

## Methods

### Experimental methods

Experiments were conducted in the Coherent X-ray Imaging (CXI) hutch at the Linac Coherent Light Source[56]. We utilize a pulse generation scheme similar to that used previously[36], utilizing the split undulator method[57]. Two collinear X-ray pulses ($\approx 60$ μJ/pulse, 10 fs) were generated in a series of undulators from a single electron bunch with a chicane delay controlling the experimental time delay ($\tau$) between the two pulses. As shown in Fig. 1 a, the electron bunch is inefficiently lased in the first half of the undulators to produce the 7.20 keV pump pulse. The remaining electron energy is utilized in the second half of the undulators to produce the lower energy 7.06 keV probe pulse. Data shown here are restricted to $\tau = 0$ fs during the overlap of the pump and probe pulses on the sample. The spectrometer calibration was performed by inserting a channel cut monochromator upstream at the XPP endstation[58]. The monochromator was tuned to multiple energies in the 7.06 keV region which showed up as valleys, missing spectral frequencies, in the resolved probe spectrum on the ePix 10 K detector[59].

We measure the transmission of the probe pulse as a function of pump fluence, achieved by measuring two focal conditions of the X-ray beams on the sample. In the first condition, we position the sample jet at the focus of the two beams. The second condition is achieved with the sample positioned upstream ($\Delta z = +2$ mm) to reduce the fluence of both pulses by increasing the spot size (-10 Rayleigh lengths from focus)[60]. As the response to this technique is linear with respect to the fluence of each pulse, small perturbations in the focal area create a quartic drop in signal strength. Calculations estimating the signal strength with this experimental design are detailed in Supplementary Note 1.

The two focal conditions serve as "pump on" and "pump off" conditions as neither pulse can be uniquely blocked or eliminated without drastically impacting the energy, intensity, and temporal profile of the other pulse. The fact that both pulses are generated from the same electron bunch creates an intrinsic link between the character and intensity of the two pulses. Future experiments will take advantage of a non-collinear generation geometry, generating pulses from distinct electron bunches, enabling direct chopping of the pump pulses either by mechanical means or through electron bunch control. Data are collected at 120 Hz, alternating 36,000 shots (5 min) in each focal condition. Total shot counts for each spectrum are provided in Supplementary Table 1.

The complexes, potassium ferrocyanide ($K_4[Fe^{II}(CN)_6]$) and potassium ferricyanide ($K_3[Fe^{III}(CN)_6]$), were purchased from Sigma Aldrich and used without further purification. Aqueous 500 mM

solutions were prepared by dissolving these complexes in ultrapure water. An HPLC pump was used to flow the solutions through a 250 μm (inner diameter) capillary. A catcher placed below the capillary refed the pump to enable closed loop recirculation of the sample. The focus spot size was $\approx$100 nm[60,61]. Beam throughput was measured to be 3%. Assuming half of the beam is in the focal volume[62], the fluence from each pulse at the sample measured $1.1 \times 10^{18}$ W/cm$^2$. Data are further filtered based on correlated intensity measures as specified in the Supplementary Note 2.

### Computational methods

Theoretical approaches for the on-the-fly Monte Carlo simulation of the electron cascade[63–65], and TDDFT calculations of the $3p$ hole state spectral signatures[66,67] have been outlined in previous work and are described in the Supplementary Note 5. The present work builds upon previous work via the consideration of the Fe$^{III}$ electron cascade and in contrasting it with that of the Fe$^{II}$ cascade.

## Data availability

The data shown in Figs. 2 and 3 are provided as Source data files. Source data are provided with this paper.

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

## Acknowledgements
This work was supported by the U.S. Department of Energy, Office of Science, Office of Basic Energy Sciences, Chemical Sciences, Geosciences and Biosciences Division under Awards DE-SC0019277 and DE-SC0023249 (R.B.W., C.E.L.-S., B.I.P., C.M.L., and M.K.), KC-030105172685 and KC-030105180818 (E.B., N.G.), and Contracts DE-AC02-76SF00515 (E.B., R.W.S.) and DE-AC02-06CH11357 (P.J.H). R.B.W. and B.I.P. acknowledge support from the NSF Graduate Research Fellowship Program under Grant No. DGE-1762114. Use of the Linac Coherent Light Source (LCLS), SLAC National Accelerator Laboratory, is supported by the U.S. Department of Energy, Office of Science, Office of Basic Energy Sciences under Contract No. DE-AC02-76SF00515. This research benefited from computational resources provided by EMSL, a DOE Office of Science User Facility sponsored by the Office of Biological and Environmental Research and located at the Pacific Northwest National Laboratory (PNNL). PNNL is operated by Battelle Memorial Institute for the United States Department of Energy under DOE contract number DE-AC05-76RL1830. The SSRL Structural Molecular Biology Program is supported by the DOE Office of Biological and Environmental Research and by the National Institutes of Health, National Institute of General Medical Sciences (including P41GM103393 and P30GM133894). The contents of this publication are solely the responsibility of the authors and do not necessarily represent the official views of NIGMS or NIH.

## Author contributions
R.B.W., C.E.L.-S., A.A., E.B., B.I.P., D.Z., R.W.S., N.G., and M.K. designed the research and experiments. R.B.W., E.B., R.A.-M., A.A., S.B., F.D.F., T.K., U.B., R.W.S., B.I.P., C.M.L., A.L., and M.K. conducted the experiment at the LCLS (onsite and remotely). R.B.W. analyzed the experimental data. R.B.W., C.E.L.-S., and N.G. performed the TDDFT calculations. P.H. performed the electron cascade calculations. R.B.W., N.G., and M.K. interpreted the results. R.B.W. and M.K. wrote the manuscript with input from all authors.

## Competing interests
The authors declare no competing interests.
