## [Peer Review File · Nature Communications]

Revealing core-valence interactions in solution with femtosecond X-ray pump X-ray probe spectroscopyReviewers' Comments:

Reviewer #1:

Remarks to the Author:

This is an important contribution in the field of ultrafast X-ray spectroscopy. It represents an important milestone on the way to multicolour X-ray spectroscopy and in particular, towards multidimensional spectroscopies. The results are neat and very well presented and I recommend publication without changes.

Reviewer #2:

Remarks to the Author:

This article reports on an ultrafast two-color X-ray pump X-ray probe transient absorption experiment performed in solution. As a proof of concept, it is applied to study the correlated interactions of valence 3d with 3p and deeper lying electrons in solvated ferro- and ferricyanide complexes. Overall, the article provides a comprehensive analysis of the experimental setup and results obtained. The authors have effectively demonstrated the capabilities of this technique, and discussed it in relation to other related techniques, such as Kbeta XES which also provides insight into 3p-3d interactions. This study demonstrates the scientific opportunities possible with the continued development of multicolor multi-pulse X-ray spectroscopy.

The importance and general interest of this work means that I recommended it is published in Nature Communications.